# Photonic water dynamically responsive to external stimuli

Koki Sano[1,2], Youn Soo Kim[1], Yasuhiro Ishida[2], Yasuo Ebina[3], Takayoshi Sasaki[3], Takaaki Hikima[4] & Takuzo Aida[1,2]

Fluids that contain ordered nanostructures with periodic distances in the visible-wavelength range, anomalously exhibit structural colours that can be rapidly modulated by external stimuli. Indeed, some fish can dynamically change colour by modulating the periodic distance of crystalline guanine sheets cofacially oriented in their fluid cytoplasm. Here we report that a dilute aqueous colloidal dispersion of negatively charged titanate nanosheets exhibits structural colours. In this 'photonic water', the nanosheets spontaneously adopt a cofacial geometry with an ultralong periodic distance of up to 675 nm due to a strong electrostatic repulsion. Consequently, the photonic water can even reflect near-infrared light up to 1,750 nm. The structural colour becomes more vivid in a magnetic flux that induces monodomain structural ordering of the colloidal dispersion. The reflective colour of the photonic water can be modulated over the entire visible region in response to appropriate physical or chemical stimuli.

[1] Department of Chemistry and Biotechnology, School of Engineering, The University of Tokyo, 7-3-1 Hongo, Bunkyo-ku, Tokyo 113-8656, Japan. [2] RIKEN Center for Emergent Matter Science, 2-1 Hirosawa, Wako, Saitama 351-0198, Japan. [3] National Institute for Materials Science, International Center for Materials Nanoarchitectonics, 1-1 Namiki, Tsukuba, Ibaraki 305-0044, Japan. [4] RIKEN SPring-8 Center, 1-1-1 Kouto, Sayo, Hyogo 679-5198, Japan. Correspondence and requests for materials should be addressed to Y.I. (email: y-ishida@riken.jp) or to T.A. (email: aida@macro.t.u-tokyo.ac.jp).

Unilamellar titanate(IV) nanosheet (TiNS; Fig. 1a)[1–5] is a two-dimensional (2D) electrolyte that consists exclusively of surface atoms that form a single-crystal-like 2D array. This inorganic nanosheet is ultrathin (0.75 nm) with an exceptionally high aspect ratio ($\sim 10^4$). It is characterized by a high density of negative charges ($1.5 \, \text{C m}^{-2}$) that are surrounded by a cloud of quaternary ammonium counterions ($Q^+$, tetramethylammonium; Fig. 1b) to form an electrical double layer. Like bulk titania, TiNS is stable in water, behaves as a photocatalyst[3,4] and exhibits a high refractive index of $>2.0$ (ref. 5). When an aqueous dispersion of TiNSs is placed in a strong magnetic field, the randomly oriented nanosheets become cofacially aligned to one another with a uniform separation that maximizes their electrostatic repulsion[6]. This anomalous orientation is caused by a particular diamagnetic susceptibility of TiNS, which forces TiNSs to align orthogonally to lines of magnetic flux[7,8]. No other metal oxide nanosheets are known to display this peculiar magnetic orientation[9,10].

During our studies, we attempted to size-fractionate TiNSs by centrifugation of their aqueous dispersion. Although this trial was unsuccessful, we found, to our considerable surprise, that the precipitate produced by the centrifugation process exhibited a vivid colour when redispersed in water (Supplementary Movie 1), suggesting the intriguing possibility that this fluid contained a periodic structure that selectively reflected visible light[11]. In fact, as highlighted here, we confirmed that TiNSs in the resulting dispersion spontaneously align in a cofacial manner with a uniformly large separation of up to 675 nm. Similar large-scale periodic structures are known to form in noncentrifuged dispersions of TiNSs, but only in the presence of a magnetic field and, even in this oriented state, the nanosheets are separated from one another by, at most, 50 nm, so that the dispersion is colourless[7]. TiNSs are obtained by exfoliation of layered protonic titanate with tetramethylammonium hydroxide ($Q^+OH^-$)[1,2]. The spontaneous structural coloration described above allowed us to notice that before the centrifugation, the intrinsically large electrostatic repulsion between cofacial TiNSs observed in our earlier samples is attenuated by a proportion of $Q^+OH^-$ that is not consumed during the exfoliation (Fig. 1c, right)[12]. Centrifugation removes this ionic contaminant, so that the dispersion displays a structural colour (Fig. 1d). This aqueous dispersion, a sort of photonic liquid crystal with an ultrahigh water content ($>99.5$ vol%), may be called 'photonic water'. It displays two apparently contradictory features: fluidity and order. When external stimuli are applied, photonic water rapidly changes colour through alteration of the periodic distance and orientation of the TiNSs (Fig. 1e).

## Results

**Preparation and characterization of photonic water.** Photonic water (Fig. 2a) is readily prepared by the following typical procedure (Fig. 2b). A TiNS dispersion with a TiNS content [TiNS] of 0.13 vol% (Fig. 2c, i) is centrifuged at 20,000$g$ for 1 h. The resultant precipitate is diluted to a [TiNS] of 0.13 vol%. Repetition of this sequence of procedures decreases the free $Q^+$ ion from 13 to about 0.1 mM (Supplementary Fig. 1), at which point the TiNS dispersion begins to exhibit purple coloration originating from a periodic cofacial structure of TiNSs (Fig. 2c, ii).

As shown in Fig. 2c,d, ii, the resultant photonic water shows notable colour unevenness owing to its randomly oriented polydomain structure. However, when a 10 T magnet field is applied from the viewing direction, the purple colour of the photonic water rapidly becomes more vivid and homogeneous, owing to magnetically induced monodomain structural ordering (Fig. 2c, iii). It is of particular interest that this crystal-like

unidirectional orientation of TiNSs develops over a range of several centimetres, so that the nanosheets reflect light in the same direction, as evidenced by the absence of any detectable birefringence under crossed Nicols (Fig. 2d, iii and Supplementary Fig. 2). Because of the additional contribution of the high refractive index of TiNS ($>2.0$, ref. 5), the reflection spectrum of the monodomain dispersion is considerably sharper and more intense than that of the polydomain version (Fig. 2e), and it also shows higher-order peaks, up to the third order (Supplementary Fig. 3). These trends indicate a high geometrical integrity of the crystal-like unidirectional orientation of TiNSs. As confirmed by the highly anisotropic pattern in 2D small-angle X-ray scattering (2D SAXS; Fig. 2f)[13], the degree of orientation is remarkably high, with an order parameter of 0.98 (ref. 14).

**Broad-range colour modulation of photonic water.** By changing [TiNS] from 0.50 to 0.09 vol%, the structural colour of the photonic water is readily modulated over an extraordinarily wide spectral range from the ultraviolet (370 nm) to the visible, and even to the near-infrared (1,750 nm) regions (Fig. 3a–c), where the spectral peak remains quite sharp (Fig. 3b). This colour-modulation range (370–1,750 nm) is the widest that has been reported for a photonic material (Fig. 3c)[15–27]. Although photonic materials capable of reflecting near-infrared radiation are indispensable for modern telecommunications systems, such materials are difficult to synthesize[22]. According to Bragg's law, a reflection wavelength of 1,750 nm (Fig. 3b, right-hand end) corresponds to a plane-to-plane TiNS separation of 675 nm. Such an ultralong separation, which is 900 times greater than the thickness of TiNS (0.75 nm), is surprising, and it has never been reported for a colloidal system[23–29]. We consider that this particular long-period structure is formed as a result of the large surface potential of TiNS arising from its high charge density ($-79$ mV) and the low concentration of contaminant electrolyte achieved by extensive deionization (a free $Q^+$ concentration of 0.12 mM at [TiNS] = 0.09 vol%). Figure 3d shows the potential for a pair of cofacial TiNSs as calculated from the Derjaguin–Landau–Verwey–Overbeek (DLVO) theory for 2D colloids[12,30] for the case of [TiNS] = 0.09 vol%. We can find a potential minimum at a TiNS distance of 670 nm, in close agreement with the observed value of 675 nm (Methods). The theory predicts that the coexisting $Q^+$ ions should screen electrostatic repulsions between cofacial TiNSs, resulting in a shrinkage of the plane-to-plane TiNS separation. In fact, the structural colour of the photonic water shows a blue shift when $Q^+$ ions are added (Supplementary Fig. 4). The critical effect of deionization on the dispersion profile of TiNSs has long been overlooked as a result of the presumption that exfoliation of layered protonic titanate should require a stoichiometric amount of $Q^+OH^-$ (Fig. 1c, left)[1,2]. However, our systematic study revealed that only a half the added $Q^+$ ions are used in replacing $H^+$ ions (Supplementary Fig. 1b,c) while the other half remain unused (Fig. 1c, right) and screen electrostatic repulsions between the cofacial TiNSs[12], as described above.

**Structural integrity of photonic water.** Next, we revisit the problem of magnetically induced monodomain structural ordering of TiNSs in their deionized dispersions by investigating whether such dispersions contain a free volume. As shown in Fig. 4b, when [TiNS] is varied from 0.09 to 0.50 vol%, the plane-to-plane TiNS separation changes in rough proportion to [TiNS]$^{-1}$, as expected for the case in which the dispersion adopts an ideal monodomain structure without any free volume (red broken line in Fig. 4b; expanded monodomain). However, the plane-to-plane TiNS separation before the deionization is

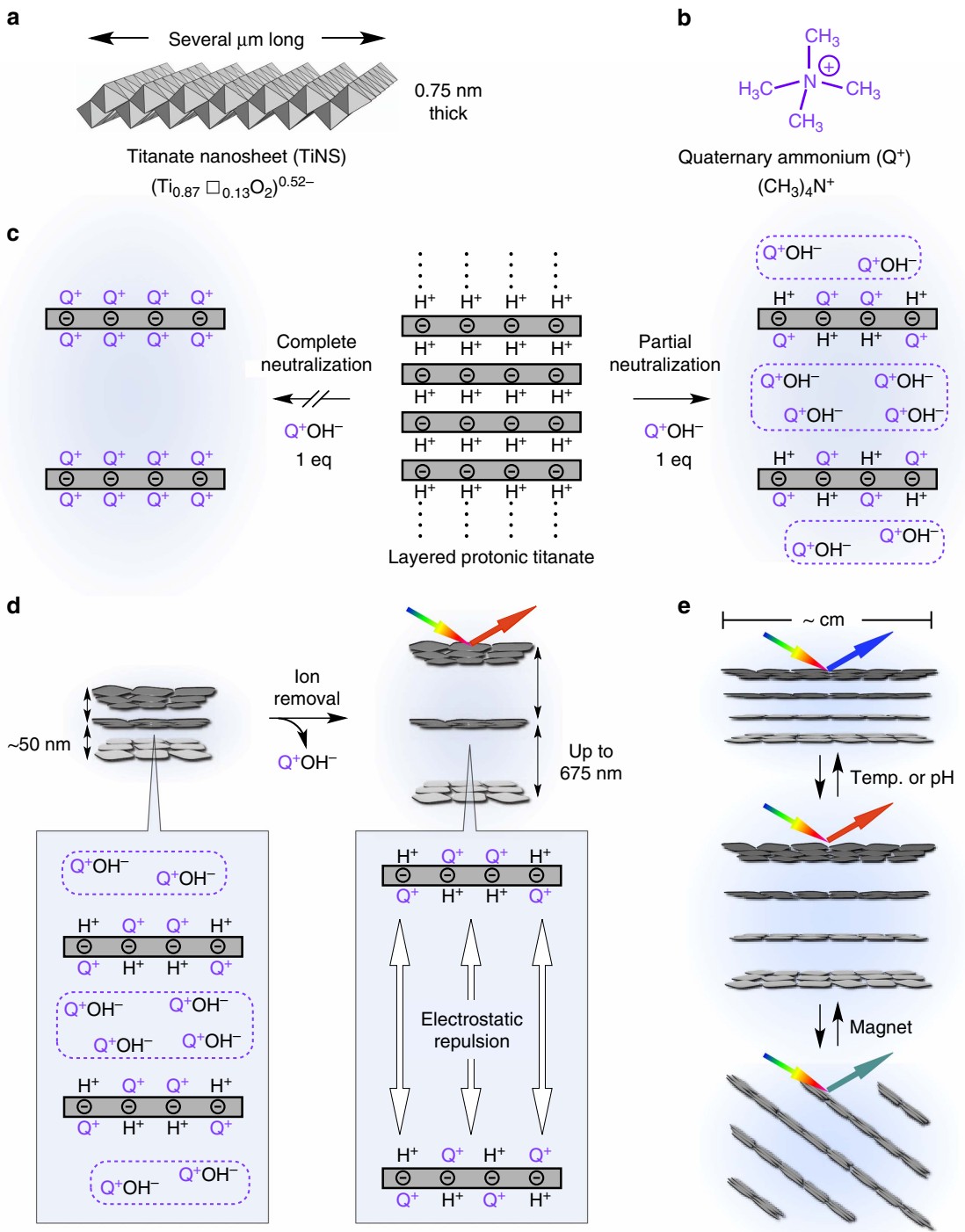

**Figure 1 | Photonic water with cofacially oriented titanate nanosheets.** (**a,b**) Schematic illustration of the structure of a negatively charged unilamellar TiNS (**a**), with quaternary ammonium (Q$^+$; tetramethylammonium) counterions (**b**). In **a**, □ indicates a vacant site. (**c**) Exfoliation of TiNSs from crystals of the layered protonic titanate by treatment with aqueous Q$^+$OH$^-$. States of ions at complete (left) or partial (right) neutralization. (**d**) Enhancement of the electrostatic repulsion between TiNSs by removal of the free ions (Q$^+$OH$^-$) that screen the electrostatic repulsion, leading to sufficient expansion of the plane-to-plane distance between cofacial TiNSs to permit reflection of visible light (up to 675 nm). (**e**) Colour modulation of the photonic water by tuning the periodic distance (upper) or the direction (lower) of TiNSs in response to external stimuli.

barely dependent on [TiNS]$^{-1}$, suggesting that the nondeionized dispersion of TiNSs might contain a large free volume (Fig. 4a; contracted polydomain). In fact, the TiNS separation at [TiNS] = 0.13 vol%, for example, is 40 nm, as determined by SAXS, which is 10 times smaller than the expected value for an ideal monodomain structure. This observation indicates that the free volume amounts to as much as 90% of the total volume of the dispersion. This large free volume originates from the presence of the ionic contaminant, which induces significant contraction of the periodic cofacial structure of TiNSs, as described above[12]. Accordingly, the geometrical relaxation of oriented TiNSs is markedly different in deionized and nondeionized dispersions.

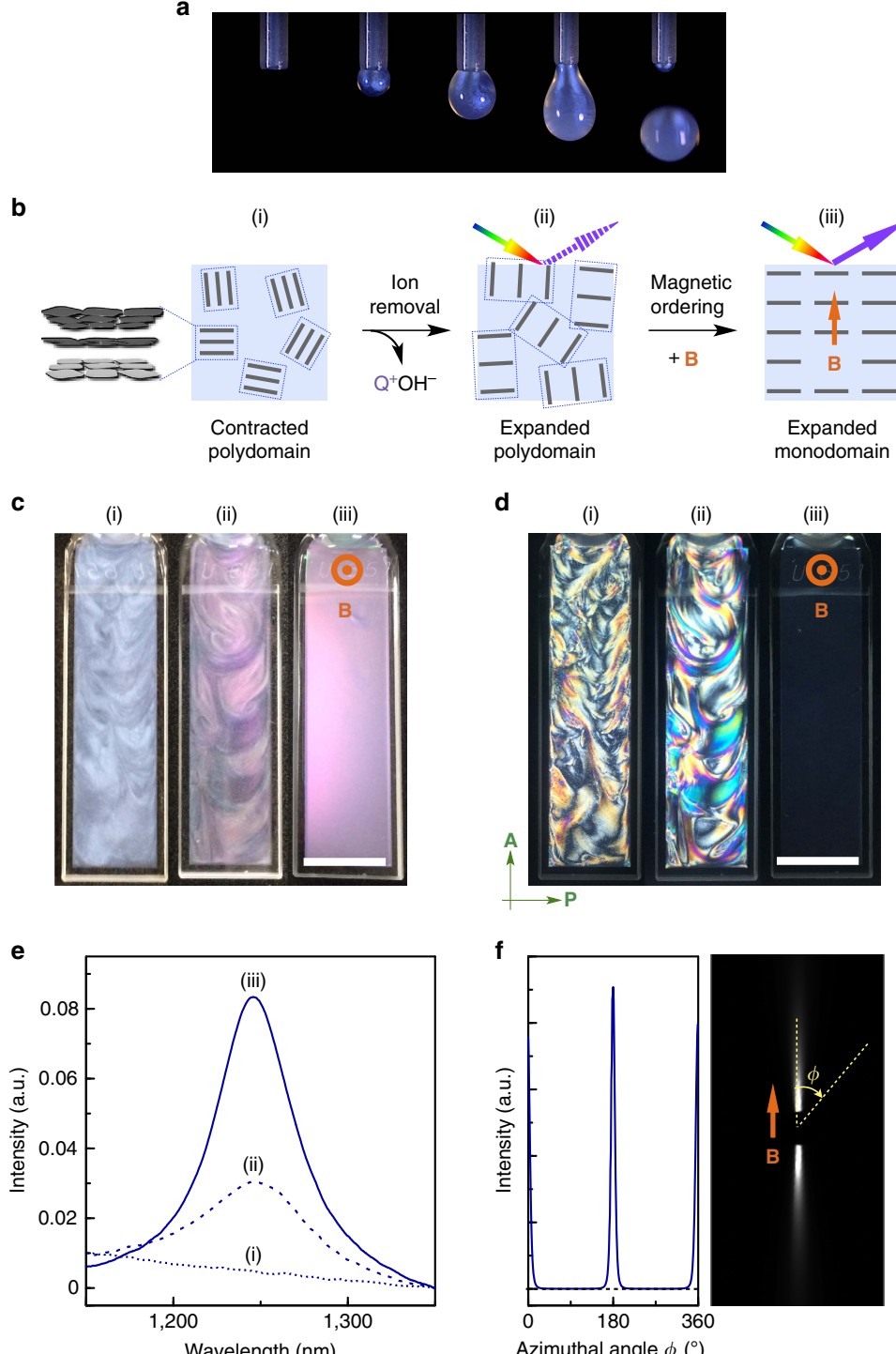

**Figure 2 | Preparation and characterization of magnetically oriented photonic water.** (**a**) Pictures of a dropping photonic water ([TiNS] = 0.45 vol%) from a thin capillary. (**b–e**) Schematic illustration (**b**), optical images (**c**), polarized optical images (**d**) and reflection spectra (**e**) at 25 °C of an aqueous TiNS dispersion (0.13 vol%) in the nondeionized (i), deionized (ii) and magnetically oriented (iii) states. In iii, a 10 T magnetic flux was applied parallel to the viewing direction. The scale bars in **c** and **d**, 1 cm. (**f**) 2D SAXS image (right) and the corresponding azimuthal angle plot (left) of the magnetically oriented photonic water ([TiNS] = 0.13 vol%) at 25 °C. Before the SAXS measurement, the sample was gelled by *in situ* polymerization of water-soluble acrylic monomers (8.0 wt%). The incident X-ray beam was directed parallel to the magnetically oriented TiNS plane. In **c–f**, the sample thickness was 1 mm.

For example, when the magnetic field is removed, the deionized system at [TiNS] = 0.13 vol% shows essentially no geometrical relaxation during 50 h; the reflection intensity at 620 nm in a 1 mm-thick quartz cuvette changes slowly with a half-life longer than 400 h (Fig. 4d)[31]. Such an excellent relaxation tolerance is maintained up to 60 °C (Supplementary Fig. 5). In sharp contrast, the nondeionized TiNS dispersion, as a reference, undergoes complete geometrical relaxation within 5 min (Fig. 4c).

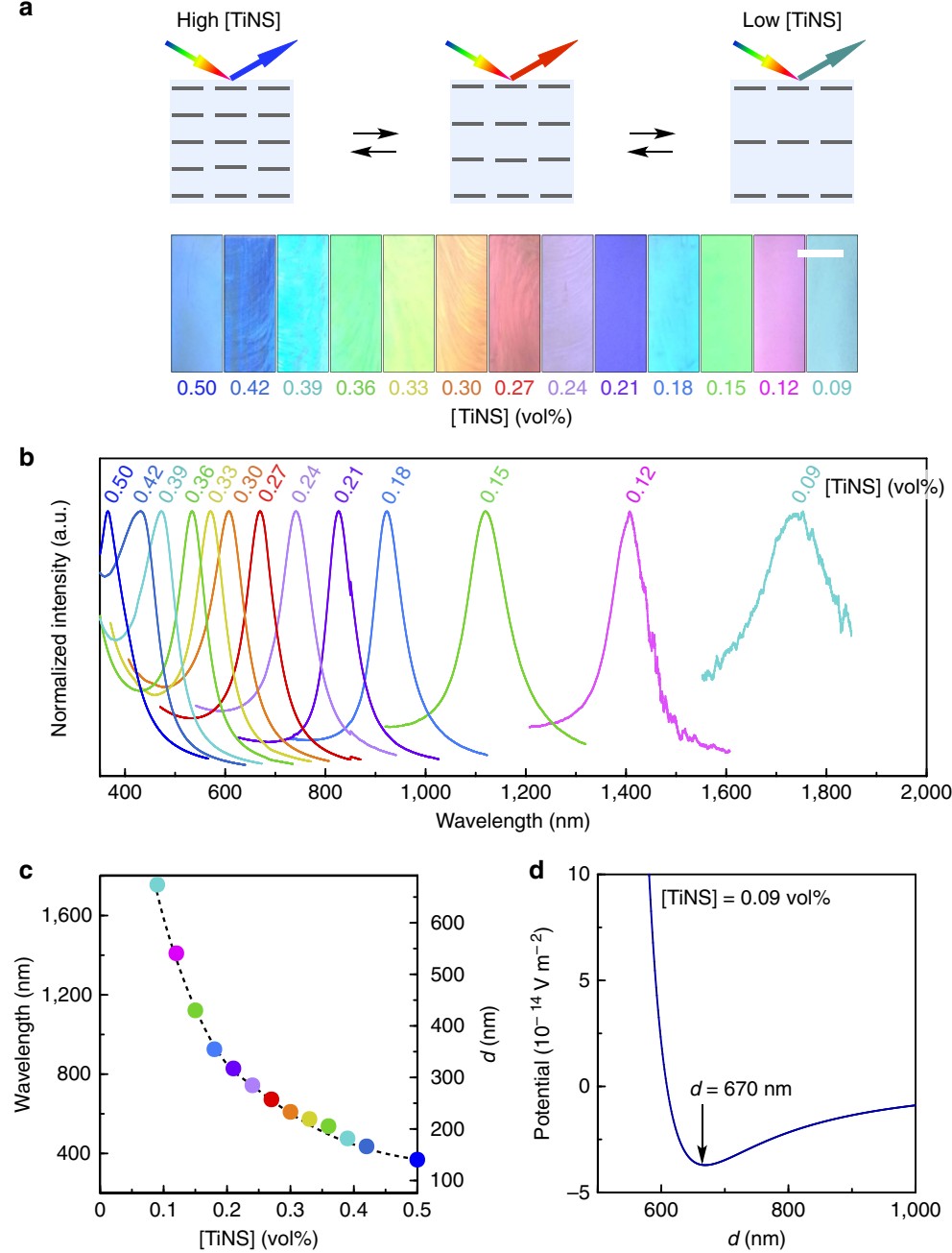

**Figure 3 | Broad-range modulation of the structural colour of photonic water.** (**a**–**c**) Optical images (**a**), normalized reflection spectra (**b**) and primary peak wavelengths (**c**) at 0 °C of magnetically oriented photonic water with various TiNS contents (0.09–0.50 vol%). The scale bar in **a**, 5 mm. (**d**) Potentials of a pair of cofacial TiNSs (0.09 vol%) as a function of the plane-to-plane distance calculated by the DLVO theory.

**Stimuli-responsive colour modulation of photonic water.** In nature, some animals such as fish rapidly modulate their colours in response to external stimuli[32–34]. For this superb function, they use soft photonic materials consisting of cofacially oriented crystalline guanine sheets in a fluidic cytoplasm. These sheets are capable of rapid dynamic changes in their periodic distance (for example, in blue damselfish[32,33]) or their direction (for example, in neon tetras[34]) in response to stimuli. No conventional photonic material has previously been reported to show such dynamic structural colour changes[15–21]. However, we found that our photonic water can show rapid changes in its structural colour. For example, when the photonic water ([TiNS] = 0.30 vol%) is heated and cooled between 0 and 80 °C, its colour changes as its reflection wavelength rapidly and reversibly shifts between 600 and 480 nm (Fig. 5a), which corresponds to the changes in the TiNS separation between 230 and 185 nm. Meanwhile, the reflection peak becomes slightly broadened on heating, due to a thermal fluctuation of the structural order of colloidally dispersed TiNSs (Supplementary Fig. 6). This colour modulation occurs due to a thermoresponsive ionic-density change in TiNSs (Supplementary Fig. 7a). Namely, heating enhances the dissociation of ion pairs on the TiNS surface to generate a larger number of free $Q^+$ ions (Supplementary Fig. 7b). Then, Ti-O$^-$ moieties thus produced are protonated, thereby reducing the surface potential of TiNSs (Supplementary

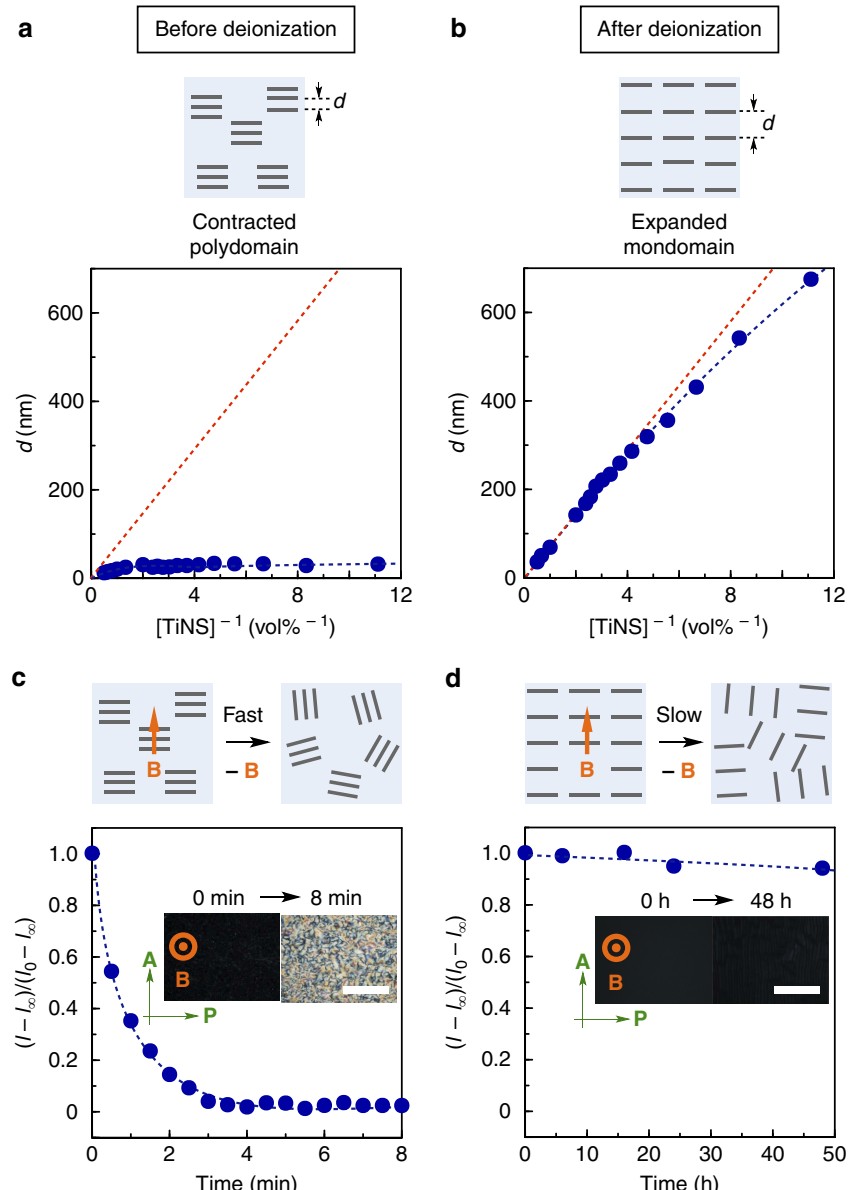

**Figure 4 | Monodomain nature of magnetically oriented photonic water.** (**a,b**) Plots of the plane-to-plane distance between cofacial TiNS ($d$) as a function of $[TiNS]^{-1}$ for the nondeionized (**a**) and deionized (**b**) aqueous TiNS dispersions (0.13 vol%). The $d$ values were estimated from the primary peak in the reflection spectrum or the SAXS profile. Blue circles: experimental results; red broken lines: calculated values for a monodomain structure with unidirectionally oriented and homogeneously distributed TiNSs ($d$ = nanosheet thickness $\times [TiNS]^{-1}$). (**c,d**) Relaxation profiles at 25 °C of magnetically ordered structures in the nondeionized (**c**) and deionized (**d**) TiNS dispersions (0.13 vol%), as monitored by the reflection intensity at 620 nm ($I$). The insets show temporal changes in the POM images of the TiNS dispersions. Scale bars, 1 mm.

Fig. 7c). These changes attenuate the electrostatic repulsion between TiNSs in water. Consequently, the TiNS separation is reduced (Supplementary Fig. 7d). We also found that this colour modulation occurs rapidly and is completed shortly within 200 ms (Supplementary Fig. 8).

An analogous colour change (420–600 nm; Fig. 5b) occurs when the direction of an applied magnetic flux ($\theta$ in Fig. 5b), which induces orientation of the cofacial TiNSs relative to the viewing direction, is varied between 0° and 90°. The change in the reflection wavelength can be well elucidated by a simple model of Bragg reflection (Supplementary Fig. 9). It is of particular interest that at $\theta = 0°$, the photonic water becomes colourless and transparent because the nanosheets are oriented parallel to the viewing direction.

Even more surprising is an extremely sharp response to pH (Fig. 5c). When titrated with hydrochloric acid, the photonic water sensitively changes its structural colour from red to green to blue with small changes in pH from 7.9 to 7.7 to 7.3, respectively, and a small pH change of 0.1 units is characterized by six different colours. The main reason for this colour change is the protonation of the oxyanionic groups on TiNSs, resulting in attenuation of their electrostatic repulsion[12].

## Discussion

In the present work, we show that photonic water provides the widest colour modulation range ever reported,

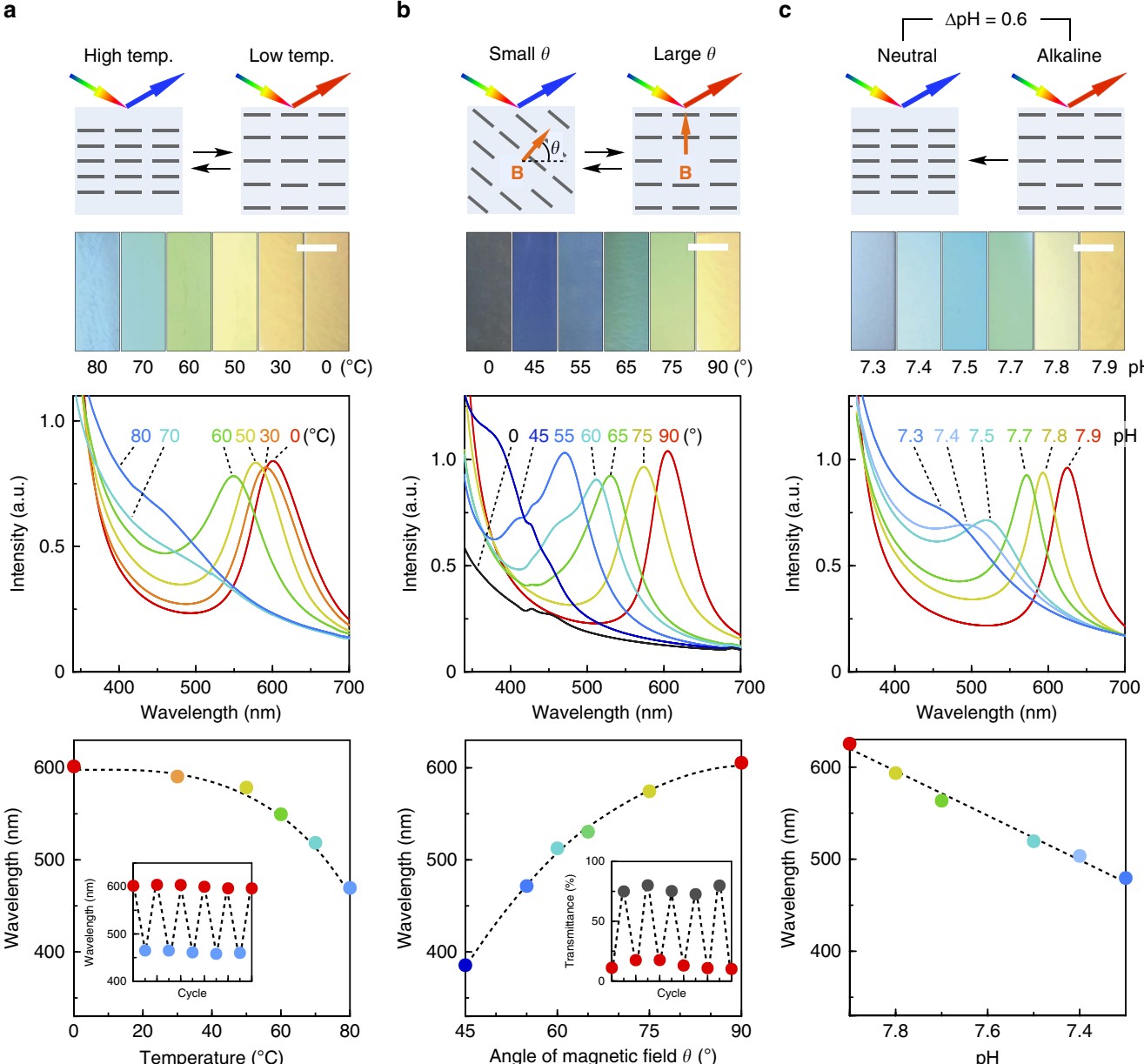

**Figure 5 | Colour modulation of photonic water in response to external stimuli.** (a–c) Optical images (upper), reflection spectra (mid) and changes in primary peak wavelength (lower) of the magnetically oriented photonic water ([TiNS] = 0.30 vol%) on changing the temperature (a), the angle of an externally applied magnetic flux (b) or the pH value (c). Scale bars, 5 mm. The insets to a and b (lower) show changes in the optical features of the photonic water on repetitive switching of the external stimuli.

including ultraviolet, visible and even near-infrared regions (370–1,750 nm) and also has the ability of quick colour modulation (for example, 120 nm/200 ms) in response to various physical or chemical stimuli. Photonic water is composed of a 'single-crystal'-like monodomain structure that develops over several centimetres, owing to particular diamagnetic susceptibility of the nanosheets. Furthermore, photonic water can visualize a very subtle pH change from 7.3 to 7.9, where the structural colour changes widely from blue (pH 7.3) to green (pH 7.7) to red (pH 7.9). Most of these findings are the result of the anomalous properties of colloidal dispersions of 2D structured electrolytes. The discovery of photonic water breathes new life into the traditional field of colloidal sciences.

## Methods

**General.** Aqueous dispersions of unilamellar TiNSs were centrifuged using a Hitachi model CF16RXII centrifuge and a Hitachi model T15A41 rotor. Redispersion of TiNSs was carried out using a HiPep laboratories model PepSyzer II shaker. Magnetic orientation of TiNSs was carried out using a JASTEC model JMTD-10T100 superconducting magnet with a vertical bore of 100 mm. Ion conductivity was measured by using a Horiba model DS-71E conductivity meter. Zeta potentials were measured by using a Malvern model Zetasizer Nano ZSP zeta potential analyser. Polarized optical microscopy was performed on a Nikon model Eclipse LV100POL optical polarizing microscope or a KEYENCE model VHX-5000 digital microscope. Photoinduced radical polymerization was conducted by using an USHIO model OPM2-502H high-pressure mercury arc lamp (500 W). Unless otherwise noted, all reagents were used as received from TCI (tetramethylammonium chloride) and Wako (hydrochloric acid, $N,N$-dimethylacrylamide and $N,N'$-methylenebis(acrylamide)). Water was obtained from a Millipore model Milli-Q integral water purification system. TiNSs were prepared according to literature methods[2].

**Deionization of aqueous dispersions of TiNSs.** An aqueous dispersion (40 ml) of TiNSs (0.13 vol%) was centrifuged at a centrifugal force of 20,000g at 25 °C for 1 h. The supernatant (~35 ml) was removed from the centrifugation tube, and the residue was allowed to be dispersed in the same amount of water. The aqueous resulting dispersion was shaken at 40 r.p.m. for 20 min at 25 °C. The sequence of these processes (Supplementary Fig. 1a) was repeated 10 times to afford a TiNS dispersion exhibiting a purple structural colour (Fig. 2c, ii).

**Magnetic orientation of TiNSs in aqueous dispersions.** A quartz cuvette (40 × 10 × 1 mm) filled with an aqueous dispersion (~400 μl) of TiNSs (0.09–0.50 vol%) was placed in the bore of a superconducting magnet (10 T). Unless otherwise noted, the cuvette was allowed to stand for 20 min such that the shortest side of the cuvette was directed parallel to the magnetic flux. For the polarized optical microscopy observation in Supplementary Fig. 2, the cuvette was allowed to stand in the superconducting magnet (10 T) for 20 min such that the longest side of the cuvette was directed parallel to the magnetic flux. For the colour modulation in Fig. 5b, the cuvette was allowed to stand in the superconducting magnet (10 T) for 20 min such that the viewing plane and the applied magnetic flux formed angles $\theta$ prescribed in Fig. 5b. The resulting dispersions were directly used for measurements.

**Reflection spectroscopic measurements.** All reflection spectroscopic measurements were performed in the absence of a magnetic field. Unless otherwise noted, reflection spectra were recorded on a JASCO model V-570 UV/VIS/NIR spectrophotometer equipped with a JASCO model ETC-717 temperature controller. An aqueous dispersion of TiNSs (~400 μl) filled in a quartz cuvette (40 × 10 × 1 mm) was used for each measurement. For evaluating the colour modulation rate in Supplementary Fig. 8, reflection spectra were recorded on a Nikon model Eclipse LV100POL optical polarizing microscope equipped with an Ocean Optics model USB4000 spectrometer and a Mettler Toledo model FP90 Central Processor connected with a model FP 82HT hot stage. A droplet (~10 μl) of a deionized TiNS dispersion (0.30 vol%) in an ambient atmosphere at 20 °C was put onto a glass-covered hot stage set at 80 °C. The reflection spectra were recorded at every 200 ms.

**Small-angle X-ray scattering analysis.** SAXS measurements were carried out at BL45XU in the SPring-8 synchrotron radiation facility (Hyogo, Japan)[13] using a Rigaku model R-AXIS IV++ imaging plate area detector or a Dectris model Pilatus 300 K-W detectors. The scattering vector $q$ ($q = 4\pi \sin\theta/\lambda$; $2\theta$ and $\lambda$ = scattering angle and wavelength of an incident X-ray beam (1.00 Å), respectively) and position of the incident X-ray beam on the detector were calibrated using several orders of layer reflections from silver behenate ($d = 58.380$ Å). The sample-to-detector distance was 2.25 m, where acquired scattering/diffraction images were integrated along the Debye–Scherrer ring, affording the corresponding one-dimensional scattering profiles. SAXS measurements of the aqueous dispersions of TiNSs were carried out in a glass capillary (1.5 mm in diameter). For the estimation of the order parameter in Fig. 2f, the magnetically oriented photonic water was in situ hydrogelated via the following procedure[7] to prevent thermal relaxation of the magnetically ordered structure. A polystyrene cuvette (40 × 10 × 10 mm) filled with an aqueous dispersion (1.2 ml) of TiNSs (0.13 vol%), containing a mixture of N,N-dimethylacrylamide (8.0 wt%) as a monomer and N,N′-methylenebis(acrylamide) (0.08 wt%) as a crosslinker, was placed in the bore of a superconducting magnet (10 T) such that one of the 10 mm sides of the cuvette was directed parallel to the magnetic flux. After being allowed to stand at 25 °C for 20 min, the mixture was exposed to a 500 W high-pressure mercury arc light in the magnetic flux, whereupon crosslinking radical polymerization proceeded almost quantitatively within 30 min, affording a self-standing hydrogel. The hydrogel was sliced into a 1 mm-thick film just before the SAXS measurement.

**Calculation of TiNS periodic distance by using the DLVO theory.** The periodic distance ($d$) of TiNSs is calculated by using the DLVO theory.
The following parameters are used:
$e$: charge of an electron ($= 1.60 \times 10^{-19}$ C)
$k_B$: Boltzmann constant ($= 1.38 \times 10^{-23}$ J K$^{-1}$)
$\varepsilon_0$: permittivity of vacuum ($= 8.85 \times 10^{-12}$ C V$^{-1}$ m$^{-1}$)
$N_A$: Avogadro constant ($= 6.02 \times 10^{23}$ mol$^{-1}$)
$A$: Hamaker constant of TiNS ($= 1.0 \times 10^{-19}$ J)
$\delta$: thickness of TiNS ($= 0.75 \times 10^{-9}$ m).
The following parameters depend on conditions, including temperature ($T$):
$\varepsilon_r$: relative permittivity of water
$I$: concentration of free Q$^+$ ions (Supplementary Fig. 7b)
$\psi_0$: Surface potential of TiNSs (Supplementary Fig. 7c).
The following potential energies are considered:
$P_A$: potential energy of a van der Waals attractive force between cofacial TiNSs
$P_R$: potential energy of an electrostatic repulsive force between cofacial TiNSs
$P_{total}$: total potential energy ($= P_A + P_R$).

According to the DLVO theory for 2D colloids[30], these potential energies are expressed as the functions of $d$:

$$P_A = -\frac{A}{12\pi}\left\{\frac{1}{d^2} + \frac{1}{(d+2\delta)^2} - \frac{2}{(d+\delta)^2}\right\}, \quad (1)$$

$$P_R = \frac{64 N_A I k_B T}{\kappa}\left\{\tanh\left(\frac{e\psi_0}{4k_B T}\right)\right\}^2 \exp(-\kappa d) \quad \text{where } \kappa^{-1} = \sqrt{\frac{\varepsilon_r \varepsilon_0 k_B T}{2N_A e^2 I}}, \quad (2)$$

$$P_{total} = P_A + P_R = -\frac{A}{12\pi}\left\{\frac{1}{d^2} + \frac{1}{(d+2\delta)^2} - \frac{2}{(d+\delta)^2}\right\} + $$
$$\frac{64 N_A I k_B T}{\kappa}\left\{\tanh\left(\frac{e\psi_0}{4k_B T}\right)\right\}^2 \exp(-\kappa d) \quad \text{where } \kappa^{-1} = \sqrt{\frac{\varepsilon_r \varepsilon_0 k_B T}{2N_A e^2 I}}, \quad (3)$$

In principle, the secondary local minimum of $P_{total}$ gives a calculated value of $d$. However, $d$ should be lower than the value supposing a monodomain, homogeneous structure (Fig. 4b), which is calculated as $\delta \times [\text{TiNS}]^{-1}$.
In the case of a TiNS dispersion (0.09 vol%) at 0 °C, where $T = 273$ K, $\varepsilon_r = 87.7$, $I = 0.12$ mM and $\psi_0 = -79$ mV, the plot of $P_{total}$ shows a secondary local minimum at $d = 670$ nm (Fig. 3d), which is smaller than $\delta \times [\text{TiNS}]^{-1}$ of 833 nm. Therefore, the calculated $d$ value is 670 nm, which is in excellent agreement with the experimentally observed value of 675 nm.
In the case of a deionized TiNS dispersion (0.30 vol%) at 0, 25, 50, 60, 70 and 80 °C, the plots of $P_{total}$ give secondary local minimums at 293, 263, 232, 221, 200 and 181 nm, respectively (Supplementary Fig. 7d, magenta). Meanwhile, $\delta \times [\text{TiNS}]^{-1}$ is 250 nm (Supplementary Fig. 7d, green). Therefore, the calculated $d$ values at 50, 60, 70 and 80 °C are 232, 221, 200 and 182 nm, respectively, and those at 0 and 25 °C are both 250 nm. The calculated $d$ values are in excellent agreement with the experimentally observed ones (Supplementary Fig. 7d, navy).

**Data availability.** The authors declare that the data supporting the findings of this study are available within the article and its Supplementary Information files.

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

## Acknowledgements
This work was financially supported by a Grant-in-Aid for Specially Promoted Research (25000005) on 'Physically Perturbed Assembly for Tailoring High-Performance Soft Materials with Controlled Macroscopic Structural Anisotropy'. We also acknowledge the ImPACT Program of the Council for Science, Technology and Innovation (Cabinet Office, Government of Japan). K.S. thanks JSPS for a Young Scientist Fellowship and the Program for Leading Graduate Schools (MERIT). The small-angle X-ray scattering measurements were performed at BL45XU in SPring-8 with the approval of the RIKEN SPring-8 Center (proposal 20160011).

## Author contributions
K.S. designed and performed all experiments; Y.S.K., Y.I. and T.A. co-designed the experiments; Y.E. and T.S. prepared colloidally dispersed TiNSs; K.S., Y.I. and T.A. analysed the data and wrote the manuscript; T.H. supported the small-angle X-ray scattering measurements at SPring-8.

## Additional information

**Competing financial interests:** The authors declare no competing financial interests.

