## [Peer Review File · Nature Communications]

Reviewers' comments:

Reviewer #1 (Remarks to the Author):

The authors have shown convincingly that 2D nanostructures can form a large scale monodomain in water. The periodic distance is sufficiently large that reflection in the infra-red spectrum can be observed. The paper has shown with details the crucial role of removing ionic contaminants in allowing the colloidal structures to remain stable at such long distances. Although not novel (ref 12 to 14 reported using similar techniques), the authors went further by employing magnetically induced alignment to achieve their results here.

Not to quibble, but there are some questions that need to be answered:

1. Pg 4, 2nd paragraph, line 1. "...shows a notable chromatic aberration..." I am a bit confused as to what this means here. Chromatic aberration refers to the failure of an optic lens to focus all colors to a single focal plane. Perhaps the authors meant to say "polychromatic"?

2. Pg 7, 1st paragraph, line 4. Was the photonic water heated under magnetic field? This is an important point that should be clarified. Later in the paragraph, the authors claimed that color modulation was completed within 200ms, which could mean entirely differently if the magnetic field is not present while effecting this change.

3. Figure 2f. In the preparation of the samples for SAXS measurement, gelation was induced to form a solid film. It is expected that the periodic distance will shrink significantly after gelation, hence allowing SAXS to probe the aligned structure of the film. Did the authors notice any change in the large scale alignment? The spot size of the SAX is typically in the mm range. Gelation may also induce alignment on its own, especially at the surface of the cuvette. This is primarily due to the increase in effective volume fraction of the TiNS sheets during shrinkage, which drives further ordering. This is a separate phenomenon from magnetically induced alignment. Were the authors able to rule this out?

4. I am particularly impressed by the stability of the lamellar structure after removal of the magnetic field. The thermal relaxation time is probably a function of temperature. I am curious at what temperature will the thermal relaxation time begin to shorten significantly? The reported switching speed for this system is 200ms under thermal stimuli. This is a very rapid response, but I wonder if it could be faster with electric field stimulation? An electric field induced alignment will certainly make this a much more practical technology since it is not easy to produce a 10T magnetic field.

Overall, this paper has shown sufficient novelty to be published. The comments above are minor in scope and I recommend this paper be accepted once they are addressed.

Reviewer #2 (Remarks to the Author):

This paper describes in great detail the preparation and optical properties of a colloidal dispersion of Titanate nanosheets that, due to their extreme thinness, are very sensitive to a very effectively charged surface which, in turn, establishes a strong electrostatic repulsion. This is in the origin of their ordering in dispersion and since the distances involved are of the order of light wavelength they present vivid colours.

The material presents exceptional optical properties and, although the material has been known for quite some time, this paper opens a door to novel applications based on the ordering in colloidal suspension. The effects of external stimuli are very intense and make the system applicable as a detector for instance.

For the above reasons I find the paper acceptable for publication in Nature Commun. I would however

like to have some points addressed before.

While mostly experimental evidences are well described some details need attention. Data presented in Figure 5b iii, is very unnatural since typically one should provide the incidence angle (in the diffraction plane) with respect to the normal as in Bragg experiments. I advise the authors to change the angle θ for $90-\theta$ for convenience. Additionally, fitting the data to a very simple model for Bragg diffraction, taking into account the refractive index of the average medium, like eq. 1 in Adv. Mater. 9, 257 (1997) may benefit the description.

Figure 1b in SI if represented in double log axes shows a perfect power law [free Q+] $\sim 1/N_{\text{cycle}}$. This should be made explicit in revision.

The effect of pH is too relevant in my opinion to be relegated to the supplementary. It has a similar entity to the effect of temperature and a more formal explanation.

Most of the discourse is plausible and all conclusions well supported but there are a few points regarding the temperature response that need more work. Thus a few questions come to mind.

What limits the range of wavelengths?

To what degree are the nanosheets parallel to each other and what interaction guarantees it even at such large distances? What interactions are established between TNS within the planes?

Nothing is mentioned regarding the width of the reflectance peak. Can the authors provide a plausible account based on what they know? Shouldn't the temperature have an impact? How should a Brownian motion of the TNS affect the distribution of orientations and separations? Is this in agreement with available experimental results?

In fact the effect of temperature is very strong and fast. Such a notable feature is very lightly touched in the paper. The phenomena underneath are not discussed as they should. The manuscript will benefit from an explicit description of the effect and the colloidal properties that induce it.

Reviewer #3 (Remarks to the Author):

The paper contributes to the field of one-dimensional (1D) photonic liquid crystals, in particular, inorganic liquid crystals assembled from 2D nanoplatelets that can show structural coloration when the interplanar distances are of the order of the visible light wavelength. To date this class of materials has been developed for a significant number of inorganic nanosheets/platelets and factors that affect the interparticle separation are well understood [e.g.11-15 and references therein].

In this work, another member, titania nanosheets (TiNS), is added to the family and its photonic properties are studied. As it was previously described by the authors, TiNS have unique magnetic properties that allow them to orient perpendicular to the direction of the magnetic field. The interesting findings here are a very broad color modulation range and a possibility to control color of the system by changing the magnetic field direction. The latter fact, however, lacks explanation. Other external stimuli that affect color of the TiNS solutions, such as concentration of the sheets and ionic strength, are well documented for other photonic liquid crystals.

In general, the writing seems rather sketchy which, in my view, decrease scientific value of the work. For a reason, the authors do not use appropriate terms "liquid crystals" and "photonic crystals" to describe their system, but replace them by "unprecedented photonic water" (why unprecedented?). As a consequence the paper lacks necessary math formalism and modelling accepted in the field. Because of this many experimental facts, such as solution colors in the near-IR region, effects of the magnetic field direction and temperature on the color change, remain unexplained. The latter may require consideration of the hydrodynamic parameters.

In my view, the paper in its present form does not bring enough new knowledge to the field and is not

ready for publication in Nature Communications.

REVIEWERS' COMMENTS:

Reviewer #1 (Remarks to the Author):

I have read the revision and is satisfied by the authors' responses. I recommend publication of the revised manuscript.

Reviewer #2 (Remarks to the Author):

I'm fine with the current state of the manuscript.

List of Additional Experiments and Theoretical Analysis

- (1) **Supplementary Fig. 5:** Measurements of the geometrical relaxation profiles of the deionized TiNS dispersion at various temperatures
- (2) **Supplementary Fig. 6:** Evaluation of the reflection peak width of the deionized TiNS dispersion at various temperatures
- (3) **Supplementary Fig. 7b:** Measurements of the free-ion concentration in the deionized TiNS dispersion at various temperatures
- (4) **Supplementary Fig. 7d:** Comparison of the observed and calculated values for the reflection wavelength of the deionized TiNS dispersion at various temperatures
- (5) **Supplementary Fig. 9:** Comparison of the observed and calculated values for the reflection wavelength of the deionized TiNS dispersion upon changing the direction of a magnetic field
- (6) **Supplementary Section 6:** Calculation of the TiNS periodic distance by using the DLVO theory

Summary of Revisions

- (1) **page 2, abstract** ----- **Reviewer #3**
The phrase ‘unprecedented’ was removed from the manuscript.
- (2) **page 3, line 19** ----- **Reviewer #3**
The revised manuscript now contains a note for representing that photonic water may be called photonic liquid crystal.
- (3) **page 4, line 7** ----- **Reviewer #1**
The phrase ‘chromatic aberration’ was replaced with ‘color unevenness’ in the revised manuscript.
- (4) **page 6, line 19** ----- **Reviewer #1**
The revised manuscript now contains a short description for representing the temperature-dependent geometrical relaxation profile of the deionized TiNS dispersion.
- (5) **page 7, line 8** ----- **Reviewer #2**
The revised manuscript now contains a short description for representing the temperature-dependent reflection peak width of the deionized TiNS dispersion.
- (6) **page 7, line 10** ----- **Reviewer #2 / Reviewer #3**
The revised manuscript now contains a short description for explaining the thermoresponsive color modulation of the deionized TiNS dispersion using the DLVO theory.

- (7) **page 7, line 20** ----- **Reviewer #2 / Reviewer #3**
The revised manuscript now contains a short description for explaining the magneto-responsive color modulation of the deionized TiNS dispersion using a simple model of Bragg reflection.
- (8) **page 17, Fig. 5c** ----- **Reviewer #2**
Fig. 5c was transferred from Supplementary Information to the main text in the revised manuscript.
- (9) **page S3, Supplementary Section 4** --- **Reviewer #1**
The revised manuscript now contains a description of the experimental details for the reflection spectroscopic measurements.
- (10) **page S4, Supplementary Section 6** --- **Reviewer #2 / Reviewer #3**
The revised manuscript now contains a description for the periodic distances between TiNSs calculated by using the DLVO theory.
- (11) **page S6, Supplementary Fig. 1c** ----- **Reviewer #2**
The revised manuscript now contains a double log plot of Supplementary Fig. 1b.
- (12) **page S8, Supplementary Fig. 3** ----- **Reviewer #3**
The revised manuscript now contains a short description for the apparent color of the photonic water with a primary peak in the near infrared region.
- (13) **page S10, Supplementary Fig. 5** ----- **Reviewer #1**
The revised manuscript now contains data for the geometrical relaxation profiles of the deionized TiNS dispersion at various temperatures.
- (14) **page S11, Supplementary Fig. 6** ----- **Reviewer #2**
The revised manuscript now contains data for the reflection peak width of the deionized TiNS dispersion at various temperatures.
- (15) **page S12, Supplementary Fig. 7** ----- **Reviewer #2 / Reviewer #3**
The revised manuscript now contains results of comparison between the observed and calculated values for the reflection wavelength of the deionized TiNS dispersion at various temperatures.
- (16) **page S14, Supplementary Fig. 9** ----- **Reviewer #2 / Reviewer #3**
The revised manuscript now contains results of comparison between the observed and calculated values for the reflection wavelength of the deionized TiNS dispersion upon changing the direction of a magnetic field.

Answers to Comments Raised by Referees

For Reviewer #1

The authors have shown convincingly that 2D nanostructures can form a large scale monodomain in water. The periodic distance is sufficiently large that reflection in the infrared spectrum can be observed. The paper has shown with details the crucial role of removing ionic contaminants in allowing the colloidal structures to remain stable at such long distances. Although not novel (refs. 12 to 14 reported using similar techniques), the authors went further by employing magnetically induced alignment to achieve their results here.

=> We appreciate these highly encouraging comments.

Not to quibble, but there are some questions that need to be answered.

[1] Page 4, 2nd paragraph, line 1, "...shows a notable chromatic aberration..." I am a bit confused as to what this means here. Chromatic aberration refers to the failure of an optic lens to focus all colors to a single focal plane. Perhaps the authors meant to say "polychromatic"?

=> In response to this kind comment, we used the term 'color unevenness' in place of 'chromatic aberration' in the revised manuscript in the main text on page 4, line 7.

[2] Page 7, 1st paragraph, line 4. Was the photonic water heated under a magnetic field? This is an important point that should be clarified. Later in the paragraph (line 16), the authors claimed that color modulation was completed within 200 ms, which could mean entirely differently if the magnetic field is not present while effecting this change.

=> All of these measurements were conducted in the absence of a magnetic field. To clarify this point, we added details of these experiments to Supplementary Information (Supplementary Section 4).

[3] Fig. 2f. In the preparation of the samples for SAXS measurement, gelation was induced to form a solid film. It is expected that the periodic distance will shrink significantly after gelation, hence allowing SAXS to probe the aligned structure of the film. Did the authors notice any change in the large scale alignment? The spot size of the SAXS is typically in the mm range. Gelation may also induce alignment on its own, especially at the surface of the cuvette. This is primarily due to the increase in effective volume fraction of the TiNS sheets during shrinkage, which drives further ordering. This is a separate phenomenon from magnetically induced alignment. Were the authors able to rule this out?

=> The refraction spectra of the deionized TiNS dispersion before and after the gelation excluded the suggested possibility. Upon gelation, the peak wavelength shifted from 599 to 564 nm, indicating a ~6% shrinkage of the periodic distance (from 230 to 217 nm). Furthermore, the peak width was slightly increased upon gelation, indicating partial orientational disordering of the TiNSs. These results demonstrate no enhanced structural ordering upon gelation.

[4] I am particularly impressed by the stability of the lamellar structure after removal of the magnetic field. The thermal relaxation time is probably a function of temperature. I am curious at what temperature will the thermal relaxation time begin to shorten significantly.

=> For the purpose of addressing this comment, we investigated the relaxation profiles at various temperatures (Supplementary Fig. 5). The relaxation time was hardly influenced by temperature in a range of 20–60 °C, but significantly shorten when the system was heated above 70 °C. We added a short description about this issue to the main text on page 6, line 19.

[5] The reported switching speed for this system is 200 ms under thermal stimuli. This is a very rapid response, but I wonder if it could be faster with electric field stimulation? An electric field induced alignment will certainly make this a much more practical technology since it is not easy to produce a 10 T magnetic field.

=> We confirmed by a preliminary experiment that an electric field can induce anisotropic orientation of TiNSs in aqueous media. However, the geometrical integrity of the orientation achieved by the e-field stimulation was much inferior to that realized by a magnetic field.

=> Electric fields may potentially cause unfavorable redox reactions. Furthermore, samples must be thin for the applied electric field to operate effectively. In contrast, in case of using magnetic fields, there is no restriction about the dimensions of samples.

Overall, this paper has shown sufficient novelty to be published. The comments above are minor in scope and I recommend this paper be accepted once they are addressed.

=> We appreciate this encouraging conclusion. Owing to several additional experiments and a theoretical analysis as described above, we believe that we are able to address all of the points raised by the reviewers.

For Reviewer #2

This paper describes in great detail the preparation and optical properties of a colloidal dispersion of titanate nanosheets that, due to their extreme thinness, are very sensitive to a very effectively charged surface which, in turn, establishes a strong electrostatic repulsion. This is in the origin of their ordering in dispersion and since the distances involved are of the order of light wavelength they present vivid colors. The material presents exceptional optical properties and, although the material has been known for quite some time, this paper opens a door to novel applications based on the ordering in colloidal suspension. The effects of external stimuli are very intense and make the system applicable as a detector for instance. For the above reasons I find the paper acceptable for publication in *Nature Commun.*

=> We appreciate these highly encouraging comments.

I would however like to have some points addressed before. While mostly experimental evidences are well described some details need attention.

[1] Data presented in Fig. 5b iii, is very unnatural since typically one should provide the incidence angle (in the diffraction plane) with respect to the normal as in Bragg experiments. I advise the authors to change the angle θ for $90^\circ - \theta$ for convenience.

=> Please understand that θ here is defined as the grazing angle of the magnetic field with respect to the cuvette plane and coincides with the angle that appears in Bragg's law.

[2] Additionally, fitting the data to a very simple model for Bragg diffraction, taking into account the refractive index of the average medium, like eq. 1 in *Adv. Mater.* **9**, 257 (1997) may benefit the description.

=> The “magnetic-field direction”-dependent color modulation was successfully elucidated using the suggested simple model of Bragg reflection, where the experimental data were in excellent agreement with the calculated ones (Supplementary Fig. 9). We added a short description to the main text on page 7, line 20.

[3] Supplementary Fig. 1b if represented in double log axes shows a perfect power law [free Q^+] $\sim 1/N_{\text{cycle}}$. This should be made explicit in revision.

=> According to this comment, we re-plotted the data, which clearly indicated the presence of a suggested relationship between [free Q^+] and the number of cycles (Supplementary Fig. 1c).

[4] The effect of pH is too relevant in my opinion to be relegated to the supplementary. It has a similar entity to the effect of temperature and a more formal explanation.

=> According to this suggestion, we moved the data on the pH responsive color modulation to the main text as Fig. 5c.

[5] Most of the discourse is plausible and all conclusions well supported but there are a few points regarding the temperature response that need more work. Thus a few questions come to mind.

(i) What limits the range of wavelengths?

=> This color modulation occurs due to a thermoresponsive ionic-density change in TiNSs (Supplementary Fig. 7a). Namely, heating enhances the dissociation of ion pairs on the TiNS surface to generate a larger number of free Q^+ ions (Supplementary Fig. 7b). Then, $Ti-O^-$ moieties thus produced are protonated, thereby reducing the surface potential of TiNSs (Supplementary Fig. 7c). These changes attenuate the electrostatic repulsion between TiNSs in water. Consequently, the TiNS separation is reduced.

=> Such thermoresponsive color modulation is quantitatively elucidated by the DLVO theory (Supplementary Section 6 and Supplementary Fig. 7d). To clarify this point, we added a description about this issue to the main text on page 7, line 10.

(ii) To what degree are the nanosheets parallel to each other and what interaction guarantees it even at such large distances? What interactions are established between TiNS within the planes?

=> By means of 2D SAXS, the order parameter of the magnetically oriented TiNS dispersion was estimated as 0.98 (Fig. 2f). This value is in the highest class for colloidal systems.

=> According to the DLVO theory, a van der Waals attractive force and an electrostatically repulsive force work competitively between TiNSs. These forces generate a potential field whose minimum defines the TiNS separation (Supplementary Section 6).

=> In the deionized TiNS dispersion, the electrostatically repulsive force between TiNSs is exceptionally large, owing to a large surface potential of TiNSs and a low free-ion concentration. Such an enhanced repulsive force ensures the large TiNS separation (Supplementary Section 6 and Fig. 3d).

(iii) Nothing is mentioned regarding the width of the reflectance peak. Can the authors provide a plausible account based on what they know? Shouldn't the temperature have an impact? How should a Brownian motion of the TNS affect the distribution of orientations and separations? Is this in agreement with available experimental results?

=> When the full width at half maximum of the reflection peak was plotted against the applied temperature, one can recognize a clear trend that the peak becomes slightly wider as the temperature turns to be higher (Supplementary Fig. 6). This result suggests a thermal fluctuation of the structural order of TiNSs due to the enhanced Brownian motion of TiNSs

upon elevating the temperature. We added a short description about this issue to the main text on page 7, line 8.

(iv) In fact the effect of temperature is very strong and fast. Such a notable feature is very lightly touched in the paper. The phenomena underneath are not discussed as they should. The manuscript will benefit from an explicit description of the effect and the colloidal properties that induce it.

=> We appreciated this suggestion. In response to this comment, we strengthened the novelty and uniqueness of this work more explicitly in the revised manuscript.

For Reviewer #3

The paper contributes to the field of one-dimensional (1D) photonic liquid crystals, in particular, inorganic liquid crystals assembled from 2D nanoplatelets that can show structural coloration when the interplanar distances are of the order of the visible light wavelength. To date this class of materials has been developed for a significant number of inorganic nanosheets/platelets and factors that affect the interparticle separation are well understood (*e.g.* refs. 11–15 and references therein).

[1] In this work, another member, titania nanosheets (TiNS), is added to the family and its photonic properties are studied. As it was previously described by the authors, TiNSs have unique magnetic properties that allow them to orient perpendicular to the direction of the magnetic field. The interesting findings here are a very broad color modulation range and a possibility to control color of the system by changing the magnetic field direction. The latter fact, however, lacks explanation.

=> The “magnetic-field direction”-dependent color modulation was successfully elucidated using a simple model of Bragg reflection, where the experimental data were in excellent agreement with the calculated ones (Supplementary Fig. 9). We added a short description to the main text on page 7, line 20.

[2] In general, the writing seems rather sketchy which, in my view, decrease scientific value of the work. For a reason, the authors do not use appropriate terms “liquid crystals” and “photonic crystals” to describe their system, but replace them by “unprecedented photonic water” (why unprecedented?).

=> We appreciate this suggestion. We removed the phrase “unprecedented” from the main text. Furthermore, we added a short note to the main text on page 3, line 19 in order to describe that “photonic water” may be called photonic liquid crystal. Nevertheless, we did not use “photonic liquid crystal” in order to emphasize that the water content of our aqueous dispersion of TiNSs is larger than 99.5 vol%.

[3] As a consequence the paper lacks necessary math formalism and modeling accepted in the field. Because of this many experimental facts, the following (i)–(iii) remain unexplained:

(i) Solution colors in the near-IR region.

=> When the TiNS content is lower than 0.21 vol%, their dispersion exhibits a primary reflection at the near-IR region. At the same time, secondary and tertiary reflections emerge in a visible region and determine the apparent color of the dispersion (see Supplementary Fig. 3). We added a short description about this issue to the caption for Supplementary Fig. 3.

(ii) Effects of the magnetic field direction on the color change

=> Please see our answer to the comment [1].

(iii) Effects of temperature on the color change, which may require consideration of the hydrodynamic parameters.

=> This color modulation occurs due to a thermoresponsive ionic-density change in TiNSs (Supplementary Fig. 7a). Namely, heating enhances the dissociation of ion pairs on the TiNS surface to generate a larger number of free Q^+ ions (Supplementary Fig. 7b). Then, $Ti-O^-$ moieties thus produced are protonated, thereby reducing the surface potential of TiNSs (Supplementary Fig. 7c). These changes attenuate the electrostatic repulsion between TiNSs in water. Consequently, the TiNS separation is reduced.

=> Such thermoresponsive color modulation is quantitatively elucidated by the DLVO theory (Supplementary Section 6 and Supplementary Fig. 7d). To clarify this point, we added a description about this issue to the main text on page 7, line 10.

In my view, the paper in its present form does not bring enough new knowledge to the field and is not ready for publication in *Nature Commun.*

=> Owing to the several different experiments and theoretical analysis we additionally conducted as described above for addressing the comments and criticisms raised by the reviewers, we believe that our manuscript is now considerably polished up.